# BioDiscViz: A visualization support and consensus signature selector for BioDiscML results

Sophiane Bouirdene[1], Mickael Leclercq[1], Léopold Quitté[1], Steve Bilodeau[2,3], Arnaud Droit[1]*

**1** Département de Médecine Moléculaire du CHU de Québec, Université Laval, Québec, QC, Canada,
**2** Département d'oncologie, Centre de recherche du CHU de Québec – Université Laval, Québec, Québec, Canada, **3** Centre de recherche sur le cancer de l'Université Laval, Québec, Québec, Canada

* arnaud.droit@crchudequebec.ulaval.ca

**Data Availability Statement:** BioDiscViz is directly implemented in R and is available under the GNU-GPL 3 license on Gitlab (https://gitlab.com/SBouirdene/biodiscviz.git) and online at https://

## Abstract

Machine learning (ML) algorithms are powerful tools to find complex patterns and biomarker signatures when conventional statistical methods fail to identify them. While the ML field made significant progress, state of the art methodologies to build efficient and non-overfitting models are not always applied in the literature. To this purpose, automatic programs, such as BioDiscML, were designed to identify biomarker signatures and correlated features while escaping overfitting using multiple evaluation strategies, such as cross validation, bootstrapping and repeated holdout. To further improve BioDiscML and reach a broader audience, better visualization support and flexibility in choosing the best models and signatures are needed. Thus, to provide researchers with an easily accessible and usable tool for in depth investigation of the results from BioDiscML outputs, we developed a visual interaction tool called BioDiscViz. This tool provides summaries, tables and graphics, in the form of Principal Component Analysis (PCA) plots, UMAP, t-SNE, heatmaps and boxplots for the best model and the correlated features. Furthermore, this tool also provides visual support to extract a consensus signature from BioDiscML models using a combination of filters. BioDiscViz will be a great visual support for research using ML, hence new opportunities in this field by opening it to a broader community.

## Introduction

In recent years, new methods of Artificial Intelligence (AI) have been deployed in bioinformatics research to provide pattern classification, biomarker identification and forecast modeling using omics data. Studying biomarker signatures is an important part of the research process as they are correlated to biological functions. Machine learning and feature selection will identify multivariate associations of biomarkers (i.e., features) and detect complex hidden patterns in the data. Considering the existence of many algorithms for feature selection and classification, multiple models are often generated with different signatures, but inconsistent overlaps between signatures were observed despite equivalent performances being frequent [1].

sophiane-bouirdene.shinyapps.io/BiodiscViz_
shinyapp/. The version used for this article can be
found under the release 1.0.

**Funding:** Dr Steve Bilodeau received a grant from
the Canadian Institutes of Health Research (Grant
Number: 387762) for the broader project
encompassing BioDiscViz. We assure you that the
funders played no role in the study design, data
collection, analysis, the decision to publish, or the
preparation of the manuscript.

**Competing interests:** The authors have declared
that no competing interests exist.

Furthermore, correlated features may not be retained by the models during their optimization when avoiding redundancy of information. Indeed, selecting a "best model" and its signatures is an equilibrium between decomplexifying the model and getting all valuable biomarkers. Often, various approaches, like ensemble learning or union of overlapping features, tend to find optimized solutions but at the cost of either side of the balance.

A solution to facilitate the generation of multiple models and signatures has been proposed with an automatic ML tool, BioDiscML [2]. BioDiscML is a new generation ML tool which has been demonstrated to be highly efficient in multiple research topics involving the identification of biomarker signatures from various types of data, such as proteomics [3], transcriptomics [4] and multi-omics (metagenomics/metabolomics, metagenomics/lipidomics) [5, 6]. Furthermore, BioDiscML proposes various conditions for choosing a "best model", but this is complex to determine as some data are too heterogeneous to propose ideal decision threshold metrics. Unfortunately, this tool does not provide visualization of the signature, hence limiting a rapid view of the results. Thus, to help in these decisions, we propose a visual tool, BioDiscViz, to support the choice of consensus features within a set of trained classifiers with their corresponding signatures.

## Design and implementation

BioDiscViz is a visual Shiny application working on Windows and Unix operating systems to support BioDiscML by presenting an interactive interface and graphs to the researchers which will improve their understanding of the results. The application is based on R [7]. It uses the framework Rshiny [8] and its dependency Rshiny Dashboard [9] and requires Rstudio [10], an integrated development environment for R.

### Input

BioDiscViz takes as input a directory containing BioDiscML output in csv format and their summary results. The best model and the classification or regression results are independently accessible. Furthermore, the tool supports multiple BioDiscML outputs in the same directory and allows rapid switching between them.

### Layout

BioDiscViz's interface is divided into two parts: a sidebar on the left and the main section on the right.

Starting with the sidebar, the first item is the "input directory" button. Clicking this button opens an interface where it's possible to choose the directory containing the BioDiscML outputs. Below, there is a submit button, which runs BioDiscViz on the selected BioDiscML results and an "example button" which runs the analysis of the example data provided. The main part of the interface is divided into four sections: Short Signature, Long Signature, Attribute Distribution, and Consensus Signatures.

Once the BioDiscML results are submitted to BioDiscViz, additional options appear in the sidebar. First, a scrollable list allows the selection of a specific BioDiscML output to study if there are multiple outputs in the same directory. Then, two sliders are present to adjust the font and label sizes of the figures, which can be modified at any time by the user and force the update of the plots. Finally, a button to download an HTML report of the results, including all the figures will be somewhere.

On the main part of the application, four sections are accessible through the sidebar. Each section is divided into two to three parts, consisting of a results summary for the models, plots, and a table.

- **Short Signature:** This section represents the results obtained by the best model of bioDiscML.

- **Long Signature:** This section displays the correlated features.

- **Attribute Distribution:** This is an additional feature in the visualizer that allows to interactively visualize the most frequently used features by different classifiers that were tested. Various thresholds for the classifiers using metrics such as the Matthew's correlation coefficient and standard deviation are available. Moreover, the number of attributes to match the experimental design is determined by the user.

- **Consensus Signatures:** This section provides a representation of the different signatures called by the majority of classifiers based on user-defined parameters.

## Representations

There is a heatmap, a PCA, t-SNE, UMAP graph and a boxplot to represent the short, long and consensus signatures (Fig 1A). The heatmap was made using ComplexHeatmap [11], an user-friendly package for better representation of heatmaps. The PCA was built using FactoExtra [12] and the UMAP, Rtsne and boxplots with ggplot2 [13].

The attribute distribution is represented under the form of a UpsetR plot [14] (Fig 1B). UpsetR is a R package generating static upset plots to visualize the intersections between the different features in the different classifiers.

BioDiscViz also gives access to the summary details for the short signature and a table of the data used for the short and long signature. The table in the shiny application is an integration of the datasets. It allows users to search for specific information using a search field. If there is a particular instance of interest, it can be easily found and highlighted within the table (S1A–S1C Fig).

Considering that non-numerical features cannot be easily integrated into PCA and heatmap with other numerical values, a particularity of BioDiscViz is the transformation of categorical features into numerical ones. This form allows users to simply annotate them on the side of the heatmaps to integrate the information contained by these features into the clustering of PCA.

## Outputs

BioDiscViz also possesses different functions to facilitate use and export of the results for archiving, sharing and publication. The first one is the creation of a report of the different graphs represented in the application which takes into account the modifications carried out by the user. The second functionality is to be able to download a sub dataset containing the information for the selected features in the "attribute distribution".

The study of consensus signatures is of great interest to allow researchers to identify new molecular targets of interest. If the best model provides a vision of which useful data were selected, the model does not necessarily use the features providing the most information. We consider that the most frequently called signatures by the classifiers contain important information for our problem. Those are the signatures we call consensus signatures. As such, giving BioDiscML's models these consensus signatures, which were left out by the best model, could potentially improve the initial results obtained by the previous best model.

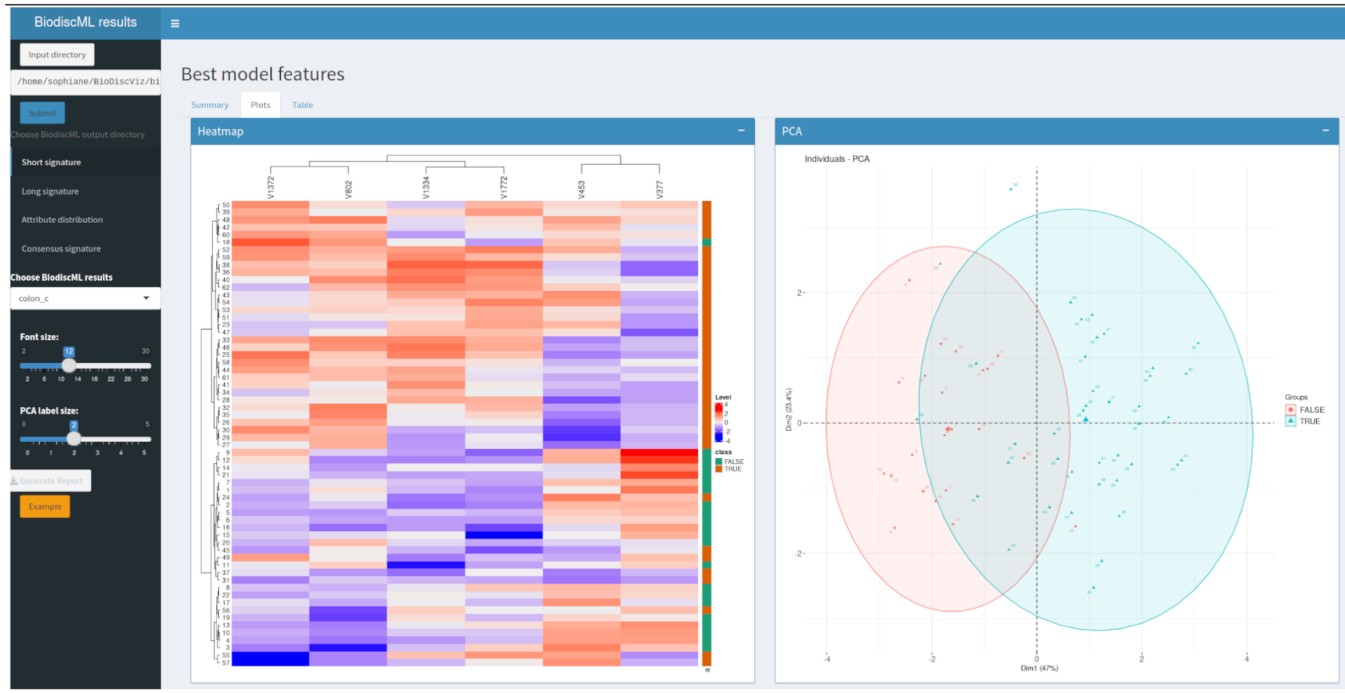

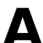

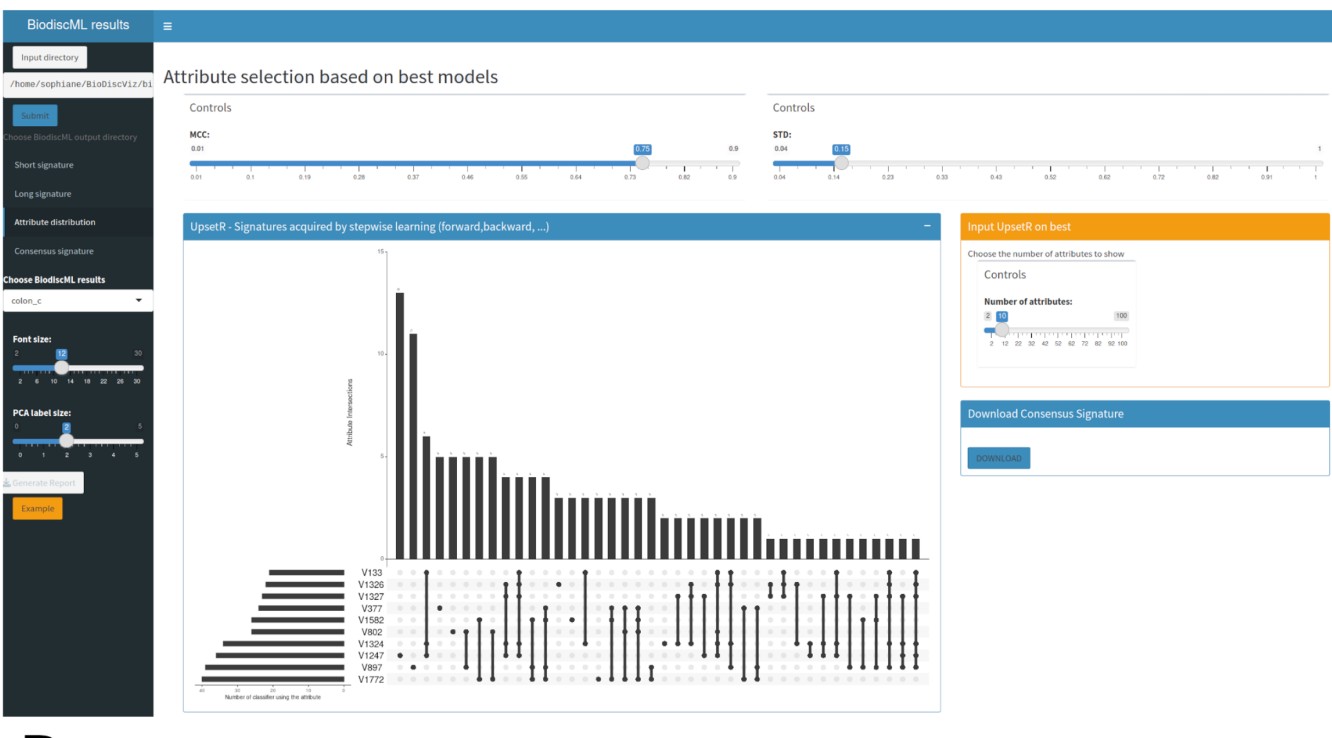

**Fig 1. Representation of the best signature and attribute distribution sections.** A. Heatmap and PCA by BiodiscViz on the best model found by BiodiscML, Kstar model, on the colon cancer dataset. We observe in cyan color the tumor tissues and in red the normal ones. B. Selection of the consensus signatures with BiodiscViz on the colon cancer dataset. Here were selected the 10 attributes most frequently called by the classifiers passing the threshold of a Matthew Correlation Coefficient $> = 0.75$ and a Standard Deviation $< = 0.15$.

## Results

To demonstrate the functionalities of BioDiscViz, we used a colon cancer dataset [15] which was used for the BioDiscML publication and which is available on BioDiscViz gitlab. This dataset contains gene expression in 40 tumor and 22 normal colon tissue samples.

### Visualize the best signature

The identification of the best signatures was studied from two perspectives. First, the signatures from the best model followed by the consensus signatures.

For signatures retrieved from the best model, different plots were generated. In this case, two classes were distinctly separated on the PCA and the heatmap (Fig 1A), showing that they provide enough information to the model to correctly predict tumor tissues and healthy tissues. Then, differential expressions of genes identified in the model were visualized using boxplots (Fig 2). Interestingly, all the signatures showed promising results as there is a clear difference for each gene between the two classes.

### Consensus signature

BioDiscML uses ensemble methods to create an association of signatures, but it does not take advantage of all generated models during its learning stage. Ensemble methods also keep all

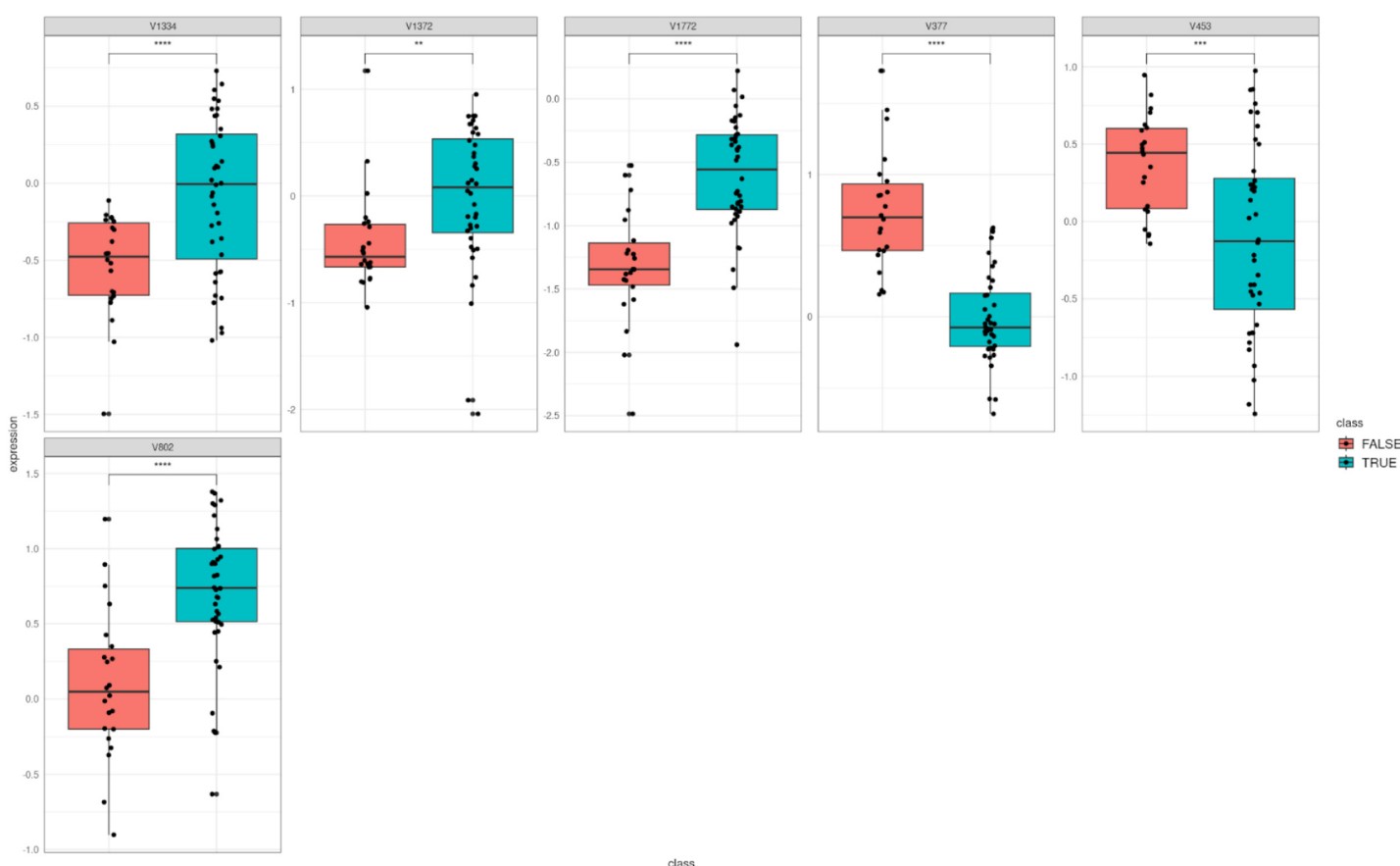

**Fig 2. Boxplot of the best model obtained by BioDiscML.** Biomarkers used by the best model selected by BioDiscML to classify the healthy and cancerous tissues.

features of the model's signatures, without any optimization, thus complexifying the model. We formulated the hypothesis that the features frequently called by the different models contained valuable information relevant to the problem at hand. The consensus signatures, referred to as such, offer a promising avenue for constructing a new signature and enhancing or streamlining the model. To delve deeper into these signatures, it is feasible to generate a dataset of consensus signatures directly using BioDiscViz. Following numerous tests, we selected the top 10 signatures from the classifiers that surpassed the MCC threshold of 0.76 (Matthew's Correlation Coefficient) and had a standard deviation of MCC (STD MCC) no greater than 0.15. (Fig 1B). The quality of the selection was assessed using the heatmap (Fig 3A) and PCA (Fig 3B), which presented a better separation between the classes than the best model identified by BioDiscML. Compared to the best model signatures, these consensus signatures consist of 3 genes overlapping with the best signature, and 7 newly added genes. To further look into these new signatures, The boxplot was used to select the genes which were differentially expressed between the healthy and cancerous tissues.

Following the identification of the consensus signature, we ran BiodiscML a second time to find an optimal machine learning classifier with the full signature, without any feature selection. The best classifier was a Kstar model with 6 attributes signature which had a MCC of 0.776 with a standard deviation across (STD) all evaluation procedures of 0.037, which is a reasonable performance considering past work on MCC evaluations [16]. Furthermore, the model had an accuracy of 0.857 Moreover, the model exhibited an accuracy of 0.857, which is comparable to, and in some cases even superior to, the results reported in the existing literature for this particular type of data [17]. With the consensus signature, we obtained a Fuzzy Lattice Reasoning model with a MCC of 0.791 (STD 0.032) which is slightly better than the previous best model (MCC increased by 1,9% and STD decreased by 15,6%).

In conclusion, our tool is able to provide visual support to BioDiscML and new insights outside of the best model by looking into the consensus signatures. Furthermore, these

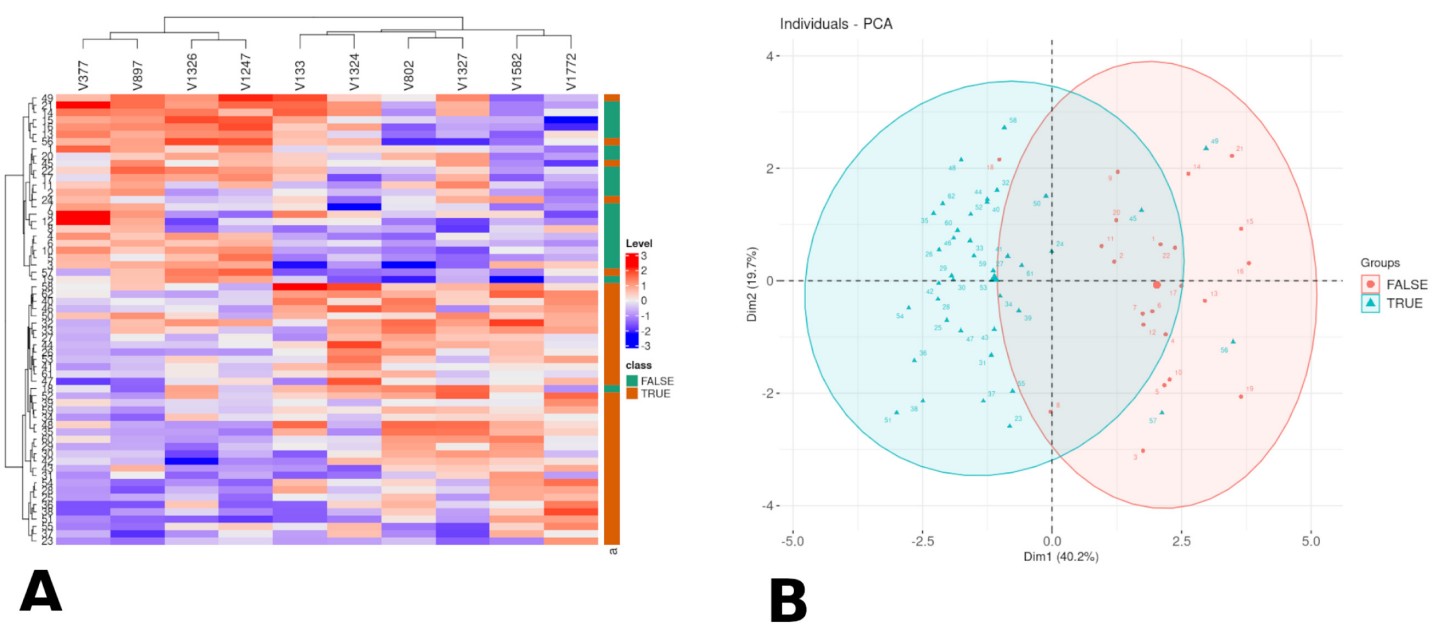

**Fig 3. Graphical representation of the 10 best consensus signatures.** A. Heatmap. B. PCA.

consensus signatures could be used to rerun BioDiscML and may enhance the quality of the model.

## Supporting information

**S1 Fig. Illustration of a research in the table.** Search for a specific instance: "12" in the table (A). The first approach involves scanning the entire table for any values that contain "12" (B). Alternatively, the user can focus on a particular column and instance and search for the one with a value of "12" (C).
(EPS)

## Author Contributions

**Conceptualization:** Sophiane Bouirdene, Mickael Leclercq, Léopold Quitté.

**Funding acquisition:** Steve Bilodeau, Arnaud Droit.

**Resources:** Mickael Leclercq, Arnaud Droit.

**Software:** Sophiane Bouirdene.

**Supervision:** Mickael Leclercq, Steve Bilodeau, Arnaud Droit.

**Validation:** Mickael Leclercq, Léopold Quitté.

**Writing – original draft:** Sophiane Bouirdene.

**Writing – review & editing:** Mickael Leclercq, Steve Bilodeau.

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
