## [Decision Letter · Decision Letter 0]

30 May 2023

PONE-D-22-29004BioDiscViz : a visualization support and consensus signature selector for BioDiscML resultsPLOS ONE

Dear Dr. Droit,

Thank you for submitting your manuscript to PLOS ONE. After careful consideration, we feel that it has merit but does not fully meet PLOS ONE’s publication criteria as it currently stands. Therefore, we invite you to submit a revised version of the manuscript that addresses the points raised during the review process. Reviewers think that the manuscript is interesting but an in-deep modifications are needed. Please take care of all theirs comments

We look forward to receiving your revised manuscript.

Kind regards,

Eduardo Andrés-León

Academic Editor

PLOS ONE

Reviewers' comments:

Reviewer's Responses to Questions

**Comments to the Author**

1. Is the manuscript technically sound, and do the data support the conclusions?

Reviewer #1: Partly

Reviewer #2: Yes

Reviewer #3: Yes

2. Has the statistical analysis been performed appropriately and rigorously? 

Reviewer #1: N/A

Reviewer #2: Yes

Reviewer #3: Yes

3. Have the authors made all data underlying the findings in their manuscript fully available?

Reviewer #1: Yes

Reviewer #2: Yes

Reviewer #3: Yes

4. Is the manuscript presented in an intelligible fashion and written in standard English?

Reviewer #1: Yes

Reviewer #2: Yes

Reviewer #3: Yes

5. Review Comments to the Author

Reviewer #1: This is not an original research paper, but rather a technical note of a visualisation extension of an already exiting software. I do not think it fits the scope of this journal and therefore, rejection is recommended

Reviewer #2: This manuscript presents BioDiscViz, an R-based visualization package for understanding and investigating the results outputted by BioDiscML. I think the tool itself is interesting and could be very useful to the users of the BioDiscML package, the presentation of the tool in this manuscript is also good but could use some improvement. For that reason, I think this manuscript is suited for publication with some editions.

Here are my suggestions for the authors to consider:

1. I was somewhat confused about the layout of BioDiscViz when I read the paper. More specifically, the authors mentioned 4 main parts, I was not sure what it was referring to, and only two of them are presented in figures 1A and 1B, so I was wondering what's on the other two parts. It's probably better to show the whole UI first, introduce each part on the UI, and focus on each individual section in later parts, or at least rearrange the existing text so it's more clear.

I actually find the "usage" part of the documentation included in the gitlab repo to be much more clear and more informative, maybe the authors could try to take some inspiration from there.

2. The authors mentioned the tool could provide an "interactive table" of the data used for the short and long signatures. I'm not quite sure what that means, but it sounds like an interesting feature. It would be great to see more details about it and probably an illustration if the authors find it fit.

3. For a visualization tool, some of the plots are a little bit crude and hard to read. One simplest changes I would suggest is increasing the font sizes of the plots, as well as the size of the scatters on the PCA plots. It would be best to make them customizable by the users.

4. Other dimension reduction plots might also be useful, like Umap, TSNE, in addition to PCA, just something to consider.

Reviewer #3: The paper presents an application, BioDiscViz, implementing different visualization techniques and exploratory tools to investigate BioDiscML outputs, that is outputs from a software implementing different Machine Learning algorithms for the identification of biomarker signatures. While I believe that the developed application has merit and that the paper is clear and well written, I have some comments for the authors.

- Currently, the application is written using the R library Shiny, but it is only hosted locally on one of the authors' gitlab pages (https://gitlab.com/SBouirdene/biodiscviz). The Shiny environment allows for free and open access web publication of Shiny applications, via https://www.shinyapps.io/. I believe the authors should make their application easily accessible on the web. In its current state, it requires for users to manually install it and run it locally in R, after having cloned the gitlab repository containing the application. This is a somehow lengthy process that could inhibit use of the application and I believe should be avoided as primary option to access it, given that a freely available platform for publishing is available. You could still leave the current option (run the app locally) as a secondary one, for more experienced users to choose.

- In relation to my previous comment, multiple packages (ComplexHeatmap - which has to be manually installed from Bioconductor, Rtools - for which the corresponding drives need to be downloaded and installed manually) are required to run the app. However, the requirement of having to install such dependencies is not stated in the instructions available in the gitlab page. This results in quite a long and multiple steps installation and deployment process for a user who does not have these library already pre-installed locally. I suggest the authors improve such aspect.

- While the illustration of usage of the app is clear on the gitlab page and on the paper, no runnable toy example is available on the app. I believe including a toy example, which would not require the upload of a BioDiscML output, would be a very useful feature of the app, that could serve better illustrating its functioning to first time users. You could consider including the same example discussed in this paper and in the gitlab page.

6. PLOS authors have the option to publish the peer review history of their article (what does this mean?). If published, this will include your full peer review and any attached files.

Reviewer #1: No

Reviewer #2: **Yes: **Xiangyun Lei

Reviewer #3: No

---

## [Author Response · Author response to Decision Letter 0]

18 Jul 2023

Reviewer #1: This is not an original research paper, but rather a technical note of a visualisation extension of an already exiting software. I do not think it fits the scope of this journal and therefore, rejection is recommended

While we respect the opinion of the reviewer, our manuscript fits the scope of PLOS ONE. According to the stated criteria, a tool must be useful to the community and demonstrate a clear advantage over existing alternatives, if applicable. For BiodiscViz, there is currently no comparable visualization tool available for machine learning results from a software like BioDiscML where many models are computed at once. Additionally, BiodiscViz represents an original contribution that significantly enhances the interpretation of machine learning results, particularly in the field of biology where researchers often possess limited knowledge in computer science. Therefore, there is a genuine need for a tool like BiodiscViz.

Reviewer #2: This manuscript presents BioDiscViz, an R-based visualization package for understanding and investigating the results outputted by BioDiscML. I think the tool itself is interesting and could be very useful to the users of the BioDiscML package, the presentation of the tool in this manuscript is also good but could use some improvement. For that reason, I think this manuscript is suited for publication with some editions.

We are pleased that the reviewer agrees with our assessment that the manuscript describes a potentially very useful tool. We have carefully considered all suggestions to enhance the research paper and the accompanying tool. We are resubmitting improved versions addressing all the comments.

 1. I was somewhat confused about the layout of BioDiscViz when I read the paper. More specifically, the authors mentioned 4 main parts, I was not sure what it was referring to, and only two of them are presented in figures 1A and 1B, so I was wondering what's on the other two parts. It's probably better to show the whole UI first, introduce each part on the UI, and focus on each individual section in later parts, or at least rearrange the existing text so it's more clear.

We thank the reviewer for pointing out this lack of clarity. We revised the Layout section and incorporated the explanation provided on GitLab to improve clarity and understanding of our tool.

 2. The authors mentioned the tool could provide an "interactive table" of the data used for the short and long signatures. I'm not quite sure what that means, but it sounds like an interesting feature. It would be great to see more details about it and probably an illustration if the authors find it fit.

The reviewer is raising an interesting question. The interactive table was not a primary feature of our tool, so we did not delve into extensive detail in the paper. The idea was to let the user access the input table without leaving the application while giving them easier means to select and search for instance or values. We are now including a brief explanation of the table and supplemented it with illustrations in the supplementary data to provide a visual demonstration of its functionality. Furthermore, as we realized that the term wasn’t the most appropriate we will just use “table” to describe it. 

 3. For a visualization tool, some of the plots are a little bit crude and hard to read. One simplest changes I would suggest is increasing the font sizes of the plots, as well as the size of the scatters on the PCA plots. It would be best to make them customizable by the users.

The comment by the reviewer relates to simplicity versus versatility of the tool. Our primary goal was to ensure that the application is highly user-friendly by minimizing the number of interactive parameters that users need to manage. However, we acknowledge that this approach could be problematic if the plots are difficult to read. Therefore, we have implemented a reactive value for the fontsize of the plots. This allows users to adjust the fontsize at any time, addressing any readability issues and providing them with greater control over their viewing experience.

 4. Other dimension reduction plots might also be useful, like Umap, TSNE, in addition to PCA, just something to consider.

We agree with the reviewer that additional dimension reduction plots would be useful. As such the Umap, and t-SNE are now included. 

Reviewer #3: The paper presents an application, BioDiscViz, implementing different visualization techniques and exploratory tools to investigate BioDiscML outputs, that is outputs from a software implementing different Machine Learning algorithms for the identification of biomarker signatures. While I believe that the developed application has merit and that the paper is clear and well written, I have some comments for the authors.

We are grateful that the reviewer appreciated the merit of our application. This new version of our manuscript addresses all the concerns.

 - Currently, the application is written using the R library Shiny, but it is only hosted locally on one of the authors' gitlab pages (https://gitlab.com/SBouirdene/biodiscviz). The Shiny environment allows for free and open access web publication of Shiny applications, via https://www.shinyapps.io/. I believe the authors should make their application easily accessible on the web. In its current state, it requires for users to manually install it and run it locally in R, after having cloned the gitlab repository containing the application. This is a somehow lengthy process that could inhibit use of the application and I believe should be avoided as primary option to access it, given that a freely available platform for publishing is available. You could still leave the current option (run the app locally) as a secondary one, for more experienced users to choose.

We thank the reviewer for the suggestion. To provide a web alternative, we now include a web option for our application to cater to the users' needs. It is now hosted in shinnyapps.io, in the free version. 

 - In relation to my previous comment, multiple packages (ComplexHeatmap - which has to be manually installed from Bioconductor, Rtools - for which the corresponding drives need to be downloaded and installed manually) are required to run the app. However, the requirement of having to install such dependencies is not stated in the instructions available in the gitlab page. This results in quite a long and multiple steps installation and deployment process for a user who does not have these library already pre-installed locally. I suggest the authors improve such aspect.

Once again, the reviewers raised an important limitation. In order to prioritize user-friendliness, we improved the tool to specifically integrate Bioconductor directly into the app. This will automatically download required packages if not already present on the user's system. Additionally, we included a reference in the readme file to guide users on downloading Rtools. Furthermore, the installation of all the libraries used in the app has been seamlessly integrated and will be automatically implemented when the user runs it. These enhancements will simplify the setup process for users and provide a smoother experience overall.

 - While the illustration of usage of the app is clear on the gitlab page and on the paper, no runnable toy example is available on the app. I believe including a toy example, which would not require the upload of a BioDiscML output, would be a very useful feature of the app, that could serve better illustrating its functioning to first time users. You could consider including the same example discussed in this paper and in the gitlab page

Initially, we included the dataset used for the research paper as an example in the "example" directory, aiming to help users become acquainted with the tool. However, it seems like our decision created confusion. We have now implemented an "example" button, which automatically loads the data specifically for this example, ensuring a better user experience. Additionally, to provide further clarity on the functioning of the app, we included a step-by-step video guide, allowing users to follow along and better understand the app's functionality. These additions will enhance user understanding and usability of the tool.

---

## [Decision Letter · Decision Letter 1]

20 Sep 2023

PONE-D-22-29004R1

BioDiscViz : a visualization support and consensus signature selector for BioDiscML results

PLOS ONE

Dear Dr. Droit,

Thank you for submitting your manuscript to PLOS ONE. After careful consideration, we have decided that your manuscript does not meet our criteria for publication and must therefore be rejected.

Specifically:

- Experiments, statistics, and other analyses are NOT performed to a high technical standard and are NOT described in sufficient detail. The visualization function can be easily achieved with R itself, and the software is NOT performed to a high technical standard.

I am sorry that we cannot be more positive on this occasion, but hope that you appreciate the reasons for this decision.

Kind regards,

Xiaoyong Sun

Academic Editor

PLOS ONE

Reviewers' comments:

Reviewer's Responses to Questions

**Comments to the Author**

1. If the authors have adequately addressed your comments raised in a previous round of review and you feel that this manuscript is now acceptable for publication, you may indicate that here to bypass the “Comments to the Author” section, enter your conflict of interest statement in the “Confidential to Editor” section, and submit your "Accept" recommendation.

Reviewer #1: All comments have been addressed

Reviewer #2: All comments have been addressed

2. Is the manuscript technically sound, and do the data support the conclusions?

Reviewer #1: Yes

Reviewer #2: Yes

3. Has the statistical analysis been performed appropriately and rigorously? 

Reviewer #1: Yes

Reviewer #2: Yes

4. Have the authors made all data underlying the findings in their manuscript fully available?

Reviewer #1: (No Response)

Reviewer #2: Yes

5. Is the manuscript presented in an intelligible fashion and written in standard English?

Reviewer #1: Yes

Reviewer #2: Yes

6. Review Comments to the Author

Reviewer #1: Thank you for addressing the comments. The manuscript has now improved and it is suitable for publication

Reviewer #2: I think the revised version properly addressed my concerns for the initial draft. I think the manuscript is more clear now and is ready to be accepted. There are few minor things for the authors to consider.

- minor typo: line 61 (might be worth to proofread again to catch other potential typos)

- it might be a good idea to include a proper license for the software, even if it's meant to be open-sourced and used by others with no restriction

- It's up to the authors, but for a software package release that might be continually maintained and updated, it might also be a good idea to create an official release (give it a version number), and archive it using Zenodo/figshare, which the authors can refer to in the paper - just so that the developer could continue working on the repo in the future, but readers would have archived version to refer to.

7. PLOS authors have the option to publish the peer review history of their article (what does this mean?). If published, this will include your full peer review and any attached files.

Reviewer #1: No

Reviewer #2: No

- - - - -

---

## [Author Response · Author response to Decision Letter 1]

17 Oct 2023

Response to editor

The visualization function included in our software, coded in R, answers the need for fast evaluation of machine learning models. While it is true that the visualization could simply be run in R, our Shinny application was designed to streamline and automate a visualization process making it accessible to researchers who may not possess coding expertise. This feature enhances the usability and accessibility of our software, aligning with the broader goal of facilitating scientific research.

Response to reviewers

We express our gratitude to the reviewers for dedicating their time to assess the second version of our manuscript. We thank them for their positive feedback. In response, particularly addressing a typo in section 6, we have made the necessary revisions. Additionally, in accordance with their recommendations, we have included a GNU GPL license for our software. Furthermore, we have initiated a release with a version number in the BioDiscViz GitLab repository, which can be accessed via the following link: https://gitlab.com/SBouirdene/biodiscviz.

---

## [Editor Report · Decision Letter 2]

9 Nov 2023

BioDiscViz : a visualization support and consensus signature selector for BioDiscML results

PONE-D-22-29004R2

Dear Dr. Droit,

We’re pleased to inform you that your manuscript has been judged scientifically suitable for publication and will be formally accepted for publication once it meets all outstanding technical requirements.

Kind regards,

Achraf El Allali, PhD

Academic Editor

PLOS ONE

Additional Editor Comments (optional):

I would have liked to see the BioDiscML fully integrated but I believe that the proposed package will have a good added value for the users who do not have the expertise to extract relevant plots and information from BioDiscML output. The authors are encouraged to maintain the web version and host it other on a paid version of shiny or on their organization's webservers.
---

## [Editor Report · Acceptance letter]

20 Nov 2023

PONE-D-22-29004R2 

BioDiscViz : a visualization support and consensus signature selector for BioDiscML results 

Dear Dr. Droit:

I'm pleased to inform you that your manuscript has been deemed suitable for publication in PLOS ONE. Congratulations! Your manuscript is now with our production department. 

Kind regards, 

on behalf of

Dr. Achraf El Allali 

Academic Editor

PLOS ONE